# Does Circular Economy Contribute to Smart Cities’ Sustainable Development?

**DOI:** 10.3390/ijerph19137627

**Published:** 2022-06-22

**Authors:** Gheorghița Dincă, Ana-Angela Milan, Maria Letiția Andronic, Anna-Maria Pasztori, Dragoș Dincă

**Affiliations:** 1Department of Finance, Accounting and Economic Theory, Faculty of Economic Sciences and Business Administration, Transilvania University of Brașov, 500036 Brasov, Romania; ana.milan@unitbv.ro (A.-A.M.); letitia.andronic@unitbv.ro (M.L.A.); anna.pasztori@unitbv.ro (A.-M.P.); 2Department of Economics 1, Faculty of Economic and Applied Economics, Bucharest University of Economic Studies, 010374 Bucharest, Romania; DINCADRAGOS15@stud.ase.ro

**Keywords:** smart city, circular economy, sustainable development, air pollution, environmental protection, recycling, renewable energy, governance performance

## Abstract

The purpose of this research paper is to investigate and identify the factors which can support the development of one characteristic of smart cities, namely, the smart environment. More specifically, the main goal is to measure the extent to which air pollution may be reduced, taking as determinants several circular economy, fiscal, and environmental factors. The Ordinary Least Squares, the Fixed Effects, and Random Effects regression models using balanced panel data were employed, over the 2011–2019 period, for 28 European states. After rigorously studying the literature, 11 indicators with a predictable impact on the exposure to air pollution were kept. According to current analysis, the most effective methods of reducing air pollution are the use of renewable energy, the investments in educating the population to reduce pollution, the proper implementation of the circular economy, and the adoption of the most suitable policies by the European Union governments. Particular attention needs to be paid to factors such as carbon dioxide-generating activities, which are significantly increasing the air pollution. Another strong value is that of providing information on the assessment of ambient air quality, and on the promotion of appropriate policies to achieve two major objectives: well-being, and sustainable cities.

## 1. Introduction

In the past years, the traditional model of “buy, consume and dispose” started to gradually be replaced by more sustainable beliefs, that slowly impose the concept of circular economy. The term circular economy (CE) may be defined as “a system that is restorative or regenerative by intention and design that can be achieved by eliminating waste through the superior design of materials, products, systems and, within this, business models” [1]. CE is thus a restructuring of whole economic and social systems, which imply a redesign of the products and services offered from their conceiving phase.

The CE concept was widely debated over the last years, its definition and conceptualization being still an open matter. Different CE definitions are to be found within peer-reviewed articles, policy papers, and consultancy reports. The variety of these definitions reveals that the CE concept has different meanings for different stakeholders. Although most of the definitions depict CE as applying the 3R (reduce-reuse-recycle) principles, some of them failed to notice the necessity of a systemic change [2].

The CE is seen as a sustainable economic system where economic growth is decoupled from resources use, through reducing consumption, and recirculation of natural resources. The notion of CE is an alternative economic framework that has gained significant momentum over the last few years. It requires the increasing attention of governments, scholars, companies, and citizens as a necessary step to achieve sustainable development (SD) at all levels of administration (local, national, global) [3]. CE practices can be applied as a “toolbox” and specific implementation approaches for achieving a sizeable number of SD targets [4]. There is a clear relationship between the CE and the sustainable development goals (SDGs), the EU practically achieving the desired SDGs by implementing initiatives that tend to make an economy more circular [5]. 

CE is a regenerative system in which resource input and waste, emission, and energy leakage are minimized by slowing, closing, and narrowing energy and material loops. This can be achieved through long-lasting design, maintenance, repairs, reuse, remanufacturing, refurbishing, and recycling. This is in contrast to a linear economy (LE). In the context of measuring the benefits of these in environmental terms, the “Lean Green” (LG) philosophy becomes relevant. It aims to produce those changes that allow a drop in the demand for materials, energy, water, and other resources, provide efficient structures, and deploy innovative technology. The current rapidly changing and highly competitive market put companies under a great pressure to adopt sustainable practices, in terms of keeping a healthy balance among economic, environmental, and social performances. The LG manufacturing approach, which combines lean practices focused on customers demand, and green practices focused on reducing the business environmental impact, has gained popularity [6]. Most papers found in the literature recognize that the two concepts of LG and CE have a great synergy for the manufacturing sector. Although with a different focus, their common element is to eliminate waste and create value in order to produce effective outcomes. Therefore, their combination seems natural [7,8,9,10]. The CE is part of the green economy, both of them actively contributing to the achievement of the SDGs, positing the need to optimize the consumption of primary resources to prevent or reduce waste and promote re-use. However, they demand more than waste management, incorporating the idea of closing resource loops whenever technically feasible. The main objective is to minimize the loss of resources that could re-enter the economic cycle through the excessive use of landfilling and incineration of waste. This requires a change in the mindset of all interested parties in society in order to promote environmentally friendly behavior.

By reusing and recycling products, the world can reduce the need for virgin materials (minerals, fossil fuels, metals, biomass) and reduce environmental impacts, ranging from greenhouse gas emissions to deforestation. The European Union (EU) Member States, The United States of America (USA), Japan and Argentina, as well as the other Shift countries, are responsible for 43% of global emissions and consume approximately a third of resources [11].

With the recognition that economic development threatens the environment, the most important institutions in the world have begun to be concerned with ensuring the sustainability of the planet. The United Nations hosted the 1972 Earth Summit where the current meaning of SD was first introduced. In 1987, the Brundtland report aimed to rethink economic development policies due to their high environmental cost [12]. In 2000, the United Nations approved the ‘Millennium Declaration’ to protect people and ensure international relations [13], and in 2015 the ‘2030 Agenda for Sustainable Development’, which consisted of a set of measures aimed to save not only humans, but the whole planet [14]. This effort to achieve sustainability of the planet is also stimulated in the European space by the European Commission’s attempt to accelerate the transition from the LE to the CE as soon as possible. The CE is considered an instrument to attain SDGs. This initiative can be found in several reports of the EU. Besides the already mentioned Declaration of the Millennium and the 2030 Agenda for Sustainable Development, there are several additional documents [15,16,17,18,19,20,21] as well: The Roadmap to a Resource Efficient Europe;A European Strategy for Plastics in a Circular Economy;A Sustainable Bioeconomy for Europe: Strengthening the Connection Between Economy, Society and the Environment;Environmental Implementation Review 2019: A Europe that protects its citizens and enhances their quality of life;Towards a Sustainable Europe by 2030;The European Green Deal;A new Circular Economy Action Plan. For a cleaner and more competitive Europe.

Furthermore, the EU has approved a series of directives and regulations between 2014 and 2019 [22,23]. Nevertheless, relevant aspects appeared in EU regulations as early as the 1970s with a clear impact on the implementation of a CE and SD in the European space [24]. 

In addition to environmental regulations, special attention should be paid to public education in order to raise citizens’ and corporations’ awareness of the measures that can and should be taken by them to reduce carbon emissions [25]. These include selective waste collection, energy saving, waste reduction in tourism, etc. In this sense, governments can play an important role in achieving sustainability by applying an efficient environmental management, using renewable energy, promoting sustainable tourism, and relying on increasing the performance of all the institutions in charge with the supervision and monitoring of all the targeted activities [26].

The term of CE has become more and more complex, by incorporating many concepts that focus on sustainability, such as: industrial ecology, eco-efficiency, waste management, renewable energy, recycling, and smart cities. Authors define CE in different manners. One definition refers to CE as a business model which is “slowing, closing, and narrowing resource loops” [27]. In other papers, CE is considered “a new frame of mind, a new perspective” [28], “a new path of industrialization” [29] or “a paradigm shift in the way things are made” [30]. Many authors appreciate CE as a system which replaces the concepts such as ‘end-of-life’ with reuse of wastes and their reduction through better-quality design of products, materials, and systems [2,31,32,33,34]. 

As the CE can hardly be captured in a short definition, one approach may be that of defining it based on the objectives that are described in the literature and that are implicit in the schools of thought that the CE paradigm emerged from [35]. The resulting hierarchy depicts a generic understanding of the values the CE follows and is meant as a first step towards an individual evaluation of these objectives.

The conclusions of a rather critical analysis of the EU’s discourse followed by a complex description of several concrete CE policies and actions adopted by the EU show a dichotomy between words and actions, with a discourse that is rather holistic, while policies focus on ‘end of pipe’ solutions and do not address the many socio-ecological implications of a circularity transition [23]. 

Many papers highlight the connection between CE, SD, and SDG’s [22,24,36,37,38]. The similarities between the concepts refer to the integration on non-economic aspects in the development equation. The key issue is assuring an SD in the flow of transferring materials and energy between nature and humans. Due to the growing population and growing consumption the pressure on environment is higher. 

For instance, in a paper published in 2022, the authors carried out and processed a large number (581) of documents on CE and SDGs and used a correlation analysis, an exploratory factor analysis, and a cluster analysis to demonstrate that SDGs do not univocally measure the concept of sustainability; there are significant relationships between CE and SDGs in the EU and the behavior of European countries is not homogeneous [22]. 

Another research paper analyzes the sustainability of the CE indicators and elaborates a multi-linear regression model with panel data for establishing the dependency of the main CE factors on EU economic growth [37]. Based on econometric modeling, during the time frame 2010–2017, the paper published in 2019 highlighted that CE generates sustainable economic growth across the EU. During the same year (2019), another paper presents the economic factors of the SD of a CE [38]. By applying econometric analysis for 27 EU countries between 2007 and 2016, the research article highlights that the extended Mankiw–Romer–Weil model is determined by resource productivity, environmental employment, recycling rate, and environmental innovation. 

The transition to a more CE is an essential contribution to the EU’s efforts to develop a sustainable, low carbon, resource-efficient and competitive economy. In this sense, a paper published in 2021 focused mainly on grouping the EU-28 countries according to their advancement towards CE using several monitoring indicators proposed by the European Commission [24]. The paper undertakes reflections in terms of CE at the macro-level, where the EU Member States are assessed for the period of 2010–2018 by the level of progress towards the CE. Based on the research, the existence of a “two-speed Europe” was identified in terms of EU countries’ advancement towards CE. The leading countries, i.e., the most advanced in pursuing operation according to CE principles, include Germany, Belgium, Spain, France, Italy, the Netherlands, and the United Kingdom (UK). The second pole accommodates EU countries in which transformation towards CE is happening at the slowest pace. This group includes mainly countries located in Central and Eastern Europe, but also the countries in the south of Europe. 

The relationship between CE and SDGs in the urban area was developed in 2021 by three economists [36]. They identified and matched the role of CE in analyzing 17 SDGs in urban scope. The ways that the CE strategy could potentially affect the SDGs, whether positively or negatively, were comprehensively evaluated. Such findings could support an equilibrium decision-making on CE promotion in cities, rather than an optimum solution to a single target under the triple bottom line (TBL) of sustainability.

As part of the efforts made for the SD of smart cities, we must also consider the practices that are based on the principle of the TBL. The TBL is one of the main systems used by businesses to assess the profits the companies are making through their corporate sustainability solutions. It is a principle launched in 1994 by Elkington [39] and discussed in different specific articles [40,41,42,43,44]. It essentially refers to the idea that companies should be encouraged to be responsible and evaluated in terms of both financial results and results that have a positive social and environmental impact.

Companies applying TBL principles must maintain as much natural order as possible, do no harm, or at least minimize environmental impact by reducing their ecological footprint, by rational management of energy consumption and non-renewable resources, and by reducing manufacturing waste, as well as making waste less toxic before disposing of it in a safe and legal way [45]. 

In this direction, it could support the LG practices, which can provide a competitive advantage (CA) to organizations and companies. These aspects are presented in recent research that addresses the link between LG’s practices, sustainability performance, and CA, as these practices can contribute to better performance in terms of sustainability and competitiveness [46]. The analysis was based on data collected through a structured questionnaire, to which 261 organizations operating in different sectors of activity (from the production sector to the services sector) from Portugal responded. A structural equation model was used by applying a set of statistical techniques to measure and study the relationships between the observable (manifest) variables and the non-observable (latent) variables. The results demonstrate a positive impact of LG’s practices on SD. Some authors investigated the CE at a municipal level. It is the case of two researchers from Germany who analyzed the extent to which the research focuses on quantifying the environmental balance of CE initiatives promoted at the municipal level [47]. The analysis scanned 300 CE initiatives reported in 83 cities around the globe and classified them into urban targets and CE strategies. Results show that 47% of the strategies focus on urban infrastructure, followed by social consumption (24%), and industries and businesses (22%). Additionally, waste management is a popular topic among the strategies, which aligns with the traditional view of CE as a waste-oriented model and a tool to create waste management policies. The topic of CE in the context of SD and the progress achieved by Romania in the last ten years at a municipal level was addressed by four Romanian economists in 2019 [48].

CE is a very much debated topic, especially in the EU, that promotes the responsible and cyclical use of resources possibly contributing to SD. In recent years, CE has been endorsed as a policy to minimize burdens to the environment and stimulate the economy. Despite different types, CE strategies can be grouped according to their attempt to preserve functions, products, components, materials, or embodied energy. In addition, these indicators can measure the LE as a reference scenario. To illustrate the classification framework, some authors selected quantitative micro-scale indicators from the literature and macro-scale indicators from the EU’s CE monitoring framework [49]. The framework illustration shows that most of the indicators focus on the preservation of materials, with strategies such as recycling.

A fixed effect panel data analysis and a second method—generalized methods of moments (GMM)—computing the Arellano–Bond dynamic panel data estimation method was used by several economists in 2020 to identify the CE’s impact on the economic growth of European countries [50]. The model includes five independent variables, such as environmental tax rate, a recycling rate of waste, private investment and jobs in a CE, patents related to recycling, and trade in recyclable raw materials. The results of both econometric models show a strong and positive correlation between CE and economic growth, highlighting the crucial role of sustainability, innovation, and investment in no-waste initiatives to promote wealth.

By adopting a fixed effects model (FEM) approach, another paper empirically examines the influence of the Chinese government’s environmental regulation, media attention, and institutional investment preference on the Enterprises’ Environmental Information Disclosure (EEID) [51]. The EEID is a major tool for companies to show the world their environmental performance, especially in terms of low-carbon development. The results demonstrate that 70% of the Chinese listed companies did not implement the EEID at all in 2020, while a lot of the other companies who did disclosure the content met a lot of problems that weakened the practicability of the EEID and had a negative impact on the government’s supervision of the environment, the demands of the public with respect to environmental protection, and the investors’ decision-making process.

The reliance on carbon-based materials and energy emission sources are highlighted as primary problems of the 21st century. If we study the datasets of Shenzhen and Shanghai’s heavy polluting listed industries (state-owned and non-state-owned) from 2014–2019, the results prove that the carbon disclosure information level is severely more negative in the case of state-owned enterprises as compared to the non-state ones [52].

Recycling represents a key factor in the CE in terms of waste management strategies, which implies material recycling. The ordinary least squares (OLS) panel regression model used in a paper published in 2018 shows that the majority of the independent variables (R&D expenditure, trade in recyclable raw materials, environmental taxes, resource productivity and domestic material consumption) are significantly influencing the main indicator of the CE, the recycling rate of municipal waste [53]. 

Municipal solid waste (MSW) management is a major social and political issue worldwide. Regional MSW management strategies are necessary. In a study from 2008, the authors measured the impacts of waste-to-electricity transformation coefficient (WETC) of incinerators and the fluctuation of unit tipping fees on the regional MSW flow/allocation of the Taipei metropolitan area from practical and economic perspectives [54]. The results are very useful for the everyday regulation of MSW administration. 

Another study, the case of the city of Johannesburg (CoJ), analyzes variables affecting solid waste generation. Population and gross domestic product (GDP) are the two compelling factors affecting MSW generation. The waste generation per capita is influenced by income level. T high income group generate on average more than the middle-or low-income groups [55]. 

There are also papers that present the current trends in the development of wastewater treatment plants (WWTP) [56]. Based on CE assumptions, challenges and barriers which prevent the implementation of the CE and the smart cities concept all feature WWTP as an important player. WWTP are to become ‘ecologically sustainable’ technological systems, and a very important nexus in smart cities.

To enable future economic development, the EU is trying to develop a sustainable and resource-efficient economy. Waste management requires a new vision and drastic improvements for a transition to a zero-waste CE. The problem is enormously complex as it involves a variety of stakeholders, demands behavioral changes, and requires a complete rethinking of the current waste management systems. In this context, scholars have analyzed policies, targets, achievements towards reaching CE targets, resource efficiency, and circular practices within energy and waste management [57,58,59,60,61,62,63,64,65,66,67,68,69]. Their studies also attempt to ascertain the linkages between CE and SD.

Cities are becoming smart not only in terms of the automation of routine functions serving individual persons, buildings, and traffic systems, but also in ways that enable us to monitor, understand, analyze, and plan the city to improve the efficiency, equity, and quality of life for its citizens in real time. Cities have to improve quality of life by creating efficiency and by better use of resources. A smart city is based on a CE, which brings about an economic, social, and environmental value. A group of eight researchers belonging to different institutions around the world developed the idea of the ‘smart cities of the future’ [70]. Their paper represents a basis for further discussion from which to argue the point that new technologies have both disruptive and synergetic effects, particularly on forms of social organization that are required both for future forms of governance and community action, but also for businesses. 

Urban planning has become essential for our very survival in the development of sustainable and green smart cities. The literature even provides a survey revealing a practical insight for anyone who wishes to discover what an eco-friendly and sustainable city-based on emerging IoT (Internet of Things) technologies is [71]. Energy-efficient practices are the key for a green sustainable city. A smart sustainable city uses information and communication technology (ICT) to improve life quality, the efficiency of urban services and operation, and competitiveness, while ensuring that it meets present and future generations’ economic, social, and environmental needs. Most of the research made in the field of ICT is intended to develop smart cities’ strategies and techniques based on collaborative IoT. 

The smart city can be understood as a sustainable city, a city performing well in economy and governance, but also in environment and living. In the specific literature, one may also find a synthesis with respect to the major actions recommended at the EU level in order to promote this vision and highlight some of the economic and institutional barriers that might be faced at all economic levels [48]. The EU recommendations associated with the trends of increasing the prices of raw materials and with the need to reduce the import dependency, so as to better ensure the energetic security and sustainability by respecting the imperatives of environmental protection, require a focus on efforts towards these directions. One of the main challenges for the future in terms of CE refers to the further development of the sectors associated with the circularity of resources and with the emergence of employment opportunities, both based on the active involvement of public and private sectors.

Another paper found in the literature aims at describing the interrelationships among smart circularity, smart product-service systems (PSS), and circular PSS concepts, and proposes a conceptual framework of smart-circular systems [72]. Firstly, the article elaborates a new understanding of smart-circular systems by articulating the base strategy smart use and extending the following circular strategies (or technical loops): maintenance, reuse, remanufacturing, and recycling. Secondly, the research outlines a critique of the state of the literature on this phenomenon and offers suggestions to guide future empirical and theoretical research in the domains of circular strategies, services, and business models. 

This paper is intended to add new empirical evidence to the existing strands of literature, by emphasizing the underlying impact of the CE indicators and SD indicators on air quality, particularly on the exposure to air pollution by particulate matter.

This paper’s originality resides in its examination of the effects of several relevant indicators on the exposure to air pollution. The explanatory variables introduced in the regression are meant to cover all areas that might produce an impact on the dependent variable. The predictors taken into consideration measure: the economic performance achieved in urban areas, the community development assistance activities, the environmental protection activities, the CE associated activities, the SD, and the governance indicators on air quality. To achieve this, an econometric model using panel data was developed. Starting from the idea that SD, CE, and SDG’s are connected concepts, the present study looks to identify and evaluate how economic performance achieved in urban areas, community development assistance activities, environmental protection activities, and good governance influence air quality. The analysis was performed for the 28 EU Member States for the 2011–2019 period, using an econometric software.

The research paper is organized as follows: the introduction defines the purpose of the research and provides a brief literature review; the second part describes the basic directives implemented by the EU states in the last years with respect to the CE; Section 3 presents the methods used to evaluate the impact of different factors on air quality; Section 4 presents empirical results; Section 5 is focused on discussing the main results obtained, while the last part of the paper offers the main conclusions and gives some pertinent directions for future analysis.

## 2. The Circular Economy Regulatory Framework at the European Union Level

At EU level, policymakers have passed a raft of regulations to cut raw materials consumption, ban single-use plastics, and reduce the overall amount of waste produced in Europe. The European Commission’s Circular Economy Action Plan helps cut waste levels and boost reuse of resources. The emphasis is on increasing products’ lifespan by making products reusable and repairable [21]. 

During the last years, the European Parliament’s Environment Committee set up a strong legislative framework for a low-carbon and zero pollution CE. For instance, one of the amendments is that of binding EU targets for 2030 to significantly reduce the EU material and consumption footprint; to stop planned obsolescence where products are designed to have short lifespan. 

In 2015, the European Commission adopted the first CE action plan [73], while three years later it adopted an entire package of measures that included the implementation of monitoring frameworks for the CE, and additional supporting legislation on plastics [16].

A new Circular Economy Action Plan [21] was adopted in March 2020, as part of a broader EU industrial strategy. It includes a sustainable product policy with prescriptions on how we make ‘things’ in order to use few materials and ensure recycling.

Other EU laws and directives of great interest in the field of CE are waste laws [74,75,76,77,78,79,80,81,82,83,84,85,86,87,88,89] whose purpose is to reach such strategic targets as: recycling all plastic packaging until 2030; reducing food waste by 50% until 2030; introducing compulsory recycling rates for different categories of waste; increasing the recycling rate of municipal waste; and reusing the raw materials which result from recycling.

The economy based on the ‘reduce-reuse-recycle’ principle is the key to reaching the targets of the European Green Deal. The EU’s economy is to become in all ways sustainable by turning climate and environmental challenges into opportunities in all policy areas [20]. The European Green Deal aims to transform the EU into a modern, resource-efficient and competitive economy. It is a structural response, a new growth strategy for Europe. 

The EU strategies and legislation cover most elements of air quality assessment and management system. Certain issues are compulsory for all Member States, while others are to be provided by national legislation. The community legislation sets the basic standards mandatory for all EU states. However, more stringent requirements can and must be laid down at national level.

The Air Quality Framework Directive [90] on ambient air quality and cleaner air for Europe creates a framework for air quality assessment. The 2008 Directive supplemented the Directive 2004/107/EC relating to arsenic, cadmium, mercury, nickel, and polycyclic aromatic hydrocarbons in ambient air [91]. 

On the one hand, at the ‘macro-level’, the Directive (EU) 2016/2284 on the reduction of national emissions of certain atmospheric pollutants lays down national emission limits for relevant pollutants. This reflects the UNECE CLRTAP Protocol (updated version of Gothenburg Protocol) to reduce acidification, eutrophication, and ground-level ozone [92]. The national emission limits must meet specified regulations by 2030. Furthermore, all Member States are required to continuously monitor emissions and to prepare emission inventories and projections.

On the other hand, at the ‘micro-level’, the Directive 2010/75/EU on Industrial emissions (IPPC) represents the major regulatory tool for certain important stationary sources (large combustion plants, waste incinerators, installations using organic solvents, installations within the scope of former IPPC Directive) [93].

The directive on the limitation of emissions of volatile organic compounds (VOC) due to use of organic solvents in certain paints and varnishes lays down maximum VOC limit values for paints, varnishes, and refinishing products and provides labeling for placing on the market such products [94]. Several other directives refer to the quality of fuels (petrol, diesel and gas-oil) [95,96]. 

In the case of mobile sources, the emissions from both road vehicles and non-road mobile machinery are regulated by two important directives: the first refers to the harmonization of Member States’ legislation regarding the measures against the emission of gaseous and particulate pollutants from compression-ignition engines for use in vehicles, and the emission of gaseous pollutants from positive-ignition engines fuelled with natural gas or liquefied petroleum gas for use in vehicles [97]; while the second reflects the harmonization of Member States with respect to the measures taken against the emission of gaseous and particulate pollutants from internal combustion engines to be installed in non-road mobile machinery [98].

In addition to the transposition of requirements laid down by the previously mentioned directives, some EU countries adopted additional legal requirements to cover stationary sources which are not regulated at the community level. 

Besides the directives which provide for air quality management explicitly, the implementation of certain directives in the field of climate change mitigation and energy efficiency may lead to air pollution reduction.

Air pollution is closely related to energy production, human health, and climate change. Since air pollution and sustainability are strongly associated with human lives, it is highly indicative to assess the impacts of air pollution on several SDGs. Air pollution impacts can be evaluated through several indicators cited under different SDG goals: good health and well-being; affordable and clean energy; sustainable cities and communities; as well as climate action.

## 3. Materials and Methods

The main purpose of this research is to support the achievement and maintenance of the smart environmental objective at an adequate level. This goal was put into practice by analyzing the influence of the main variables that are indicating progress of the CE in Europe on the air pollution phenomenon through panel data regression procedures. More precisely, the model used in the paper incorporates explanatory variables reflecting: the economic performance in urban areas, the community development assistance activities, the environmental protection activities, the CE, the SD, and the sustainable governance.

The analysis begins with the question: how effective are the CE indicators in influencing the environmental performance?

To demonstrate the empirical relationship between indicators of the CE and air pollution, the panels were built for the 2011–2019 period using the European Commission’s databases [99]. However, the information extracted from these sources was subject to statistical data availability.

The challenges faced by smart cities in their development process stem from the need to identify the optimal mechanisms to be used by city administrations so that the decision-making level is based on scientific and practical methods of analyzing existing situations and future needs of communities.

Some cities are at a more or less advanced stage in their transition towards the specific objectives of smart cities, namely: smart economy (based on innovative spirit, productivity, and labor market flexibility); smart people (characterized by the desire to learn and invest in themselves throughout their life, through creativity and flexibility); smart governance (based on participation in decision-making and transparent governance); smart mobility (characterized by local accessibility, sustainable transport systems, a developed information and communication technology infrastructure, innovation and safety); smart life (characterized by social cohesion, housing quality, health conditions, cultural facilities, and tourist attractions); and last but not least, smart environment (characterized by attractive natural conditions, environmental protection, and sustainable resource management) [100].

In a CE, waste streams are up-cycled for greater value and products are designed for disassembly, reuse, and recycling. This concept, often associated with manufacturing, can also be applied to cities, creating climate-smart hubs to save money, lower emissions, and improve living standards.

Most studies address the topic of smart cities’ development by focusing on the technological side. Researchers build their analyses by considering the technical endowment perspective in particular [101,102]. It is important to pay attention to the manner in which cities reach the desired degree of development, and to the impact on future generations. A city based on CE principles would reframe itself as a closed loop, where production of goods is linked to waste streams, where energy is created locally, and where the cities’ people, businesses, and governments build on that value to be healthier, more prosperous, and with a lower carbon footprint. Smart cities that take this path will rely only on technology-enabled solutions and data to create efficiencies and opportunities that can provide new and more impactful solutions to core smart cities systems [101]. Moreover, according to the just mentioned reference of the Public Utilities Fortnightly Association, “the core of the smart city is data, data about processes, energy, emissions, utility of resources, and behavior. A smart city can use this data to measure, and then act, in a circular way. Without the rich data a smart city delivers, a circular economy could not be implemented”. Implementing the CE in cities can bring tremendous economic, social, and environmental benefits, with the CE holding a great potential for cities. Statistics show that cities account for 85% of global GDP, and are huge collectors of materials and nutrients, accounting for 75% of natural resource consumption. Cities also produce 50% of global waste and 60–80% of greenhouse gas emissions. These are elements of the “take, make, waste” LE. As a result of their high concentration of resources, capital, and data, spread over a relatively small geographic area, cities are uniquely positioned to drive a global transition towards a CE [102].

For the present research, we used 11 independent variables, namely: mean income by degree of urbanization, official development assistance as share of gross national income, share of environmental taxes in total tax revenues, environmental protection investments of total economy, average CO_2_, trade in recyclable raw materials by waste, recycling rate of municipal waste, circular material use rate, share of renewable energy in gross final energy consumption by sector, and participation rate in education and training. All of these were analyzed in order to identify whether there is a serious correlation to the dependent variable, i.e., government effectiveness and exposure to air pollution by particulate matter.

Table 1 presents all the explanatory variables used, the main economic function they accomplish, and a brief definition with respect to their relevance to this study. Similarly, Table 2 reflects a description of the dependent variable considered in the analysis.

For a better understanding of the relevance and of the way the dependent variable is measured, Figure 1 might offer a clear comparison of the particulate matter particles between human hair, which is 50–70 microns in diameter, dust, pollen, and mold, which have less than 10 microns in diameter, on the one hand, and combustion particles, organic compounds, metals, whose diameter is extremely small: less than 2.5 μm, on the other hand.

Particulate matter contains microscopic solids or liquid droplets that can be inhaled and may even get into the bloodstream. These particles are to be found in many shapes and sizes and can be made up of hundreds of different chemicals. Most particles are the result of oxides and dioxides emitted from factories, power plants, automobiles, etc. [103].

The regression equation used to analyze the correlations among the variables described in Table 1 and Table 2 is:Expos_air_pollution = α + β_1_Mn_Urb_Inc_it_ + β_2_Develop_assist_it_ + β_3_Share_env_tax_it_ + β_4_Env_Prot_Inv_it_ + β_5_Av_CO_2it_ + β_6_Trade_Rec_Waste_it_ + β_7_Rec_Rate_Mun_Waste_it_ + β_8_Circ_Mat_Use_it_ + β_9_Share_Renew_En_it_ + β_10_Tertiary_Education_it_ + β_11_Gov_Effectiv_it_ + μ_i_ + ε_it_(1)
where:α—the constant of the regression equation (the intercept for all countries);β_1,2,…,11_—the coefficient for each explanatory variable in the regression equation (OLS, Fixed Effects Model, Random Effects Model);i—the EU country analyzed, $i=\bar{1,\ldots ,\;28}$;t—the year analyzed of the panel data time period, $t=\bar{2011,\ldots ,\;2019}$;μ_i_—the time constant individual specific effects; as proved in the results section, the random effect model assumes that the entities’ error is not correlated with the explanatory variables;ε_it_—the idiosyncratic error term (“regular” error term), which varies over countries and time.

The influences of all these variables on the dependent variable after running the model are presented in the section of empirical results. Table 3 summarizes the main descriptive characteristics of the variables used, from a statistical point of view.

The standard deviation summarizes the variability in a dataset and it illustrates the extent to which the values of each variable deviate from their average. The trade in recyclable raw materials by waste is one of the variables highlighting the heterogeneity of the 28 EU countries with respect to using the imports of recyclable raw materials. The countries recording the highest imports of recyclable raw materials by waste are Germany, with a nine-year average of 9,107,102 thousand euro/year, Italy, with an average of 4,208,801 thousand euro/year, Belgium, with an average of 4,155,033 thousand euro/year, and Spain, with an average of 3,062,393 thousand euro/year. The countries with the lowest values are Cyprus, with an average of only 399 thousand euro/year, and Malta, with an average of 460 thousand euro/year.

Another variable which is fairly heterogeneous within the 28 EU countries is represented by the mean income by degree of urbanization. The nine-year average value of almost 10,000 euro/year highlights the great differences between EU countries concerning the income in the urban area. The highest income earned in urban settings is recorded in Luxembourg, the average income for the 2011–2019 period being of 43,694 euro/year. Luxembourg is followed by Denmark, with an average of 31,818 euro/year, Ireland (28,638 euro/year), Sweden (28,284 euro/year), France (26,414 euro/year), and the Netherlands (24,627 euro/year). The countries with the average annual lowest income earned in urban settings are Romania (only 3897 euro), Bulgaria (5161 euro), and Hungary (6473 euro). 

The highest average annual investments for environmental protection during 2011–2019 were made in Germany (11,836 million euro/year), followed by France (9884 million euro/year), UK (7383 million euro/year), and Italy (5573 million euro/year). The countries with the lowest environmental protection investments of total economy are Malta (27 million euro/year), Cyprus (80 million euro/year), and Latvia (115 million euro/year). 

The standard deviation of the indicator measuring the official development assistance as share of gross national income is 0.2831%. The average of grants or loans that were undertaken by the official sector with the objective of promoting economic development and welfare in recipient EU countries was 0.42% from GNI/year in the tested period. The countries with most grants or loans that were undertaken by the official sector with the objective of promoting economic development and welfare are Sweden (with an average of 1.05% of GNI/year), Luxembourg (0.998% of GNI/year), and Denmark (0.80% of GNI/year). The countries with the lowest official development assistance as share of gross national income were Croatia (0.095 of GNI/year), Romania (0.097% of GNI/year), and Bulgaria (0.10% of GNI/year).

To analyze the dataset further and study the behavior of the chosen countries over time, panel data was used to account for individual heterogeneity. In our data set, the panel consists of an entity (country) and time (year). To control the unobserved differences among countries or changes over time, the OLS regression method allowed for the impact analysis of the relevant identified regressors on the response variable—exposure to air pollution by particulate matter. OLS is a statistical method for estimating the unknown parameters in a linear regression model by the principle of minimizing the sum of the squares of the differences between the observed response variable and those predicted by the linear function of the independent variables.

The steps of analysis of the proposed model are:Step 1. Pooled OLS Model: this presumes that there is no unique characteristic in the cross-sections (countries) and no universal effect across time. The diagnostic tests implemented are multicollinearity, variance inflation factor (VIF), and heteroscedasticity;Step 2. Fixed Effect model/LSDV Model: this presumes that there are unique features of individual cross-section that do not change during time and are not correlated with individual dependent variables (y) of cross section;Step 3: Random Effect Model: this presumes that there are some systematic random effect of individual cross-section and there are unique, time constant feature of individuals that are not correlated with the individual independent variables (x_1_, x_2_,…, x_11_);Step 4: Best Panel Model: this was established with the Hausman specification test (H_0_: Random effect is the appropriate model; H_1_: Fixed effect model is the appropriate model).

## 4. Results

To better understand the evolution of the phenomenon, Table 4 highlights, for each country and for the EU as a whole, the decrease in the degree of air pollution by particles in 2019 as compared to 2011.

Considering the final values (2019) in relation to the initial values (2011), each country recorded a favorable downward trend in terms of air pollution. Nevertheless, the rate of decrease was different from one country to another. Figure 2 further underlines this issue.

The graphic shows that the level of pollution at EU level (dial_1) has a decreasing trend, with a decrease of almost a third during the nine years, from 18.40 (particulates < 2.5 µm) to 12.60 (particulates < 2.5 µm) [40]. The European countries that have reduced the level of pollution above the EU average are Bulgaria (52.54%), Slovakia (48.31%), Hungary (45.66%), Italy (43.66%), Cyprus (42.24%), France (41.57%), Denmark (38.65%), the Netherlands (38.10%), Belgium (37.29%), Austria (36.84%), Slovenia (36.51%), Germany (36.26%), and Finland (32%). However, 15 countries have reduced pollution levels, below the EU average. A cut of less than 20% in the exposure of air pollution by particulate matter for the studied period is seen in the case of Lithuania (8.26%), Spain (8.53%), Portugal (14.95%), Romania (15.90%), Malta (16.41%), and Greece (17.06%). 

At first sight, Bulgaria made particular efforts to reduce the level of pollution by 52.54% (from 41.30 in 2011 to 19.60 in 2019). However, it still has work to do, as the EU average in 2019 was of only 12.60. From this perspective, Bulgaria has a degree of pollution 55% higher than the average, being the most polluted country during the whole period.

At the same time, although states such as Spain, Lithuania, Portugal, Ireland, and Sweden seem to have put less effort into reducing pollution, they still have a lower pollution level than the EU average.

To highlight the drivers of increasing good health and well-being, in general, as well as those of reducing air pollution, in particular, and their impact, the first step was that of developing a multiple regression model on a balanced nine-year panel data set (2011–2019), for the 28 Member States of the EU (as identified starting with 2013 when Croatia joined the EU until 2019, with the UK officially leaving the EU in 2020).

Initially, there was a group of 21 variables selected to measure the extent of their influence on the exposure to air pollution. The results showed an influence of the selected group of 77.73% (R-squared = 0.7773, Prob. > F = 0.00). However, since not all of them proved to be statistically significant (*p*-value greater than 5% or even 10%), the retrograde selection method kept only 11 factors as being relevant to the achievement of the air pollution reduction target. The 11 indicators (as seen in Table 1) proved to have an influence on the exposure to air pollution by particle matter by 67.33% (R-squared = 0.6733, Prob. > F = 0.00).

The regression analysis offers different statistical measures such as the *t*-test, which is a type of inferential statistic used as a testing tool. *T*-test looks at the *t*-statistic (*t*-Stat) which, together with the *t*-distribution values, and the degrees of freedom, allows the identification of the statistical significance of a model. Two-tail *p*-values test the hypothesis that each coefficient is different from 0. Usually, if the probability value (*p*-value) is lower than 0.05, than the predictor variable has a significant influence on the outcome variable. However, alpha may also be chosen at a threshold level of statistical significance of 0.10. Besides offering the precise *t*-Stat value and the two-tail *p*-values, Table 5 also refers to the degree of influence of each independent factor on the dependent variable, the ideal influence of these factors being strongly positive or strongly negative.

The *p*-value is the probability of obtaining results at least as extreme as the observed results of a statistical hypothesis test, assuming that the null hypothesis is correct. A smaller *p*-value means that there is stronger evidence in favor of the alternative hypothesis.

By looking to each independent variable, there are three situations that emerge:Five variables (Share_Renew_En, Tertiary_Education, Gov_Effectiv, Trade_Rec_Waste, and Develop_Assist) have negative coefficients and a significant impact on the evolution of the response variable (with *p*-value < 0.05); it means that they have a notable influence in reducing air pollution;Three variables (Av_CO_2_, Share_Env_Tax, and Rec_Rate_Mun_Waste) have positive coefficients and a significant impact on the dependent variable (with *p*-value < 0.05); it means that they have an impact on increasing air pollution;Three variables (Env_Prot_Inv, Mn_Urb_Inc, and Circ_Mat_Use) do not seem to have a significant impact on the target variable (with *p*-value > 0.05); in the ideal situation, these indicators should have had high negative values indicating reduced air pollution.

Table 6 illustrates the countries with the highest and the lowest values from the perspective of the factors that influence in a positive manner the quantity (μm) of air pollution.

To diagnose the reliability of the model, the multicollinearity test was conducted using the variance inflation factors (VIF). For the entire group of 11 independent variables, the mean VIF value was 2.88 (below 5), which means a moderate degree of multicollinearity between predictors. The higher the variation of influence factors, the less reliable the regression results are going to be. In general, a VIF above 10 indicates high correlation. Nevertheless, some authors suggest a more conservative level of even 2.5 [104]. 

Following Breusch–Pagan/Cook–Weisberg tests (with *p*-value 0.0002 < 0.05, chi2 = 14.06) and White heteroscedasticity test (with *p*-value 0.0001 < 0.05, chi2 = 133.64), the null hypothesis of homoscedasticity was rejected, the variation of errors not being constant over time. As a consequence, the Pooled OLS model was found to be unsuitable, the model showing signs of heteroscedasticity. 

In order to present a clearer picture of the evolution over time of the dependent variable, the specificities of each country and their evolution over time need to be addressed. To analyze the heterogeneity factor, two models were considered for comparison: fixed effects model (FEM) and random effects model (REM). 

FEM regression assumes that there are unique features of individual cross-sections that do not change during time and these unique features are not correlated with individual dependent variable y. REM regression implies that there are some systematic random effects of individual cross-sections and there are unique, time constant features of individuals that are not correlated with the individual independent variables x_1_, x_2_,…, x_11_. For the pooled OLS model, FEM, and REM, the results are summarized in Table 7. The table also shows some other basic tests such as the F-test, Wald test, or the values of the coefficient of determination (R-Squared), the estimates of the standard deviation of Mu (μ) and Epsilon (ε), and the link between them expressed by the interclass correlation (rho).

An important statistical test is the basic F-test, which was named after Ronald Fisher who developed F-statistic as the variance ratio in the 1920s. The Wald test (named after Abraham Wald) assesses constraints on statistical parameters based on the weighted distance between the unrestricted estimate and its hypothesized value under the null hypothesis. The highest value of the ratio between the two mentioned tests is that of 174.37 and is found in the case of REM. The coefficient of determination shows the proportion of the variation in the dependent variable that is predictable from the independent variables. For the overall model, the best results seem to also be guaranteed by REM. The last three indicators refer to the estimates of the standard deviation of the time constant individual specific effects (sigma_u), and of the idiosyncratic error term, which varies over countries and time (sigma_e). In the FEM case, 94.07% of the variance is due to differences across panels, while for REM the result is lower, of 83.08%.

Both models are well-formed, the results being guaranteed with a probability of more than 99% (for both FEM and REM, prob. > F and prob. > chi2 = 0.00).

On the one hand, FEM shows the following main results:two variables with negative coefficients (Share_Renew_En and Circ_Mat_Use) have a significant impact on reducing air pollution (particulates < 2.5 μm). An increase by 0.55% of the share of renewable energy in gross final energy consumption and an increase by 0.22% of the circular material use rate brings about a cut by one unit (micrometer) of the air pollution by particulate matter;two variables with positive coefficients (Av_CO_2_ and Mn_Urb_Inc) have a relevant influence in increasing air pollution. When the average quantity of carbon dioxide per km increases by 0.142 g or the mean urban income grows by 0.0004 euro/year, the pollution of the air goes up by one unit;seven variables (Develop-Assist, Share_Env_Tax, Env_Prot_inv, Trade_Rec_Waste, Rec_Rate_Mun_Waste, Tertiary_Education, and Gov_Effectiv) do not have an important impact of the dependent variable, their *p*-value being over 0.05. The ideal situation would have been that of recording high negative values in order to reduce air pollution.

On the other hand, REM reveals the following main results: four variables with negative coefficients (Share_Renew_En, Circ_Mat_Use, Tertiary_Education, and Gov_Effectiv) have a significant impact on the evolution of the dependent variable. An increase by 0.24% of the share of renewable energy in gross final energy consumption, by 0.17% of the circular material use, by 0.16% of the share of the population aged 25–34 who have successfully completed tertiary studies, and by 0.15% of the performance of the government contribute to a decrease by one unit of the air pollution by particulate matter;one variable with positive coefficient (Av_CO_2_) has a significant impact on the exposure to air pollution. When the average quantity of carbon dioxide per kilometer increases by 0.163 units, air pollution goes up by one micrometer;six variables (Mn_Urb_Inc, Develop_Assist, Share_Env_Tax, Env_Prot_Inv, Trade_Rec_Waste, and Rec_Rate_Mun_Waste) have a less significant impact on the measured variable (*p*-value > 0.05). In the ideal situation, these six factors should have had high negative values, helping to reduce air pollution.

Nevertheless, each model has its own limitations, and, in order to select the best model of estimation, the Hausman test was performed. The general assumptions are as follows: H_0_: Random effect model is the appropriate model;H_1_: Fixed effect model is the appropriate one.

Equation (2) shows the chi2(9) result in the econometric software:chi2(9) = (b − B)′[(V_b_ − V_B_)^−1^ (b − B),(2)

Since chi2(9) is 15.01, Prob. > chi2 = 0.0907, the alternative hypothesis can not be accepted, so that the null hypothesis seems to me more adequate.

Moreover, when performing the Breusch and Pagan Lagrangian multiplier test for RE, the results are: chibar and 2(01) = 199.31 and Prob. > chibar2 = 0.000.

Considering REM for this panel data regression implies relying on four levers of action to achieve effective pollution reduction actions: share of renewable energy in gross final energy consumption by sector, circular material use rate, tertiary educational attainment, and government effectiveness. FEM, on the other hand, brings forward only two variables: share of renewable energy in gross final energy consumption by sector, and circular material use rate.

Furthermore, Table 8 summarizes several main statistical results (*t*-test, *p*-values, coefficients of the predictors) for each robust effect model: FEM, and REM.

All in all, the Hausman test is useful to decide which model is more appropriate for the study, and would better support the proposals for future action to achieve the main objective of reducing air pollution as much as possible.

In conclusion, REM is found to be more adequate for this study. Choosing the right model for the analysis is essential for two important reasons:the mathematical models for FEM and REM are different, and the wrong model can lead to the wrong conclusion;random effects are overlooked altogether—people do not realize that it could affect their results and that pseudoreplication (lack of statistical independence in the data set) is a big problem from a statistical point of view.

The Breusch–Pagan/Cook–Weisberg test and the White test regarding heterosckedasticity proved that the null hypothesis of homoskedasticity has been rejected, the variation of errors not being constant over time, the model showing signs of heterosckedasticity. The tests for the FEM-robust and REM-robust were conducted in order to assess the proposed research assumptions and to increase the reliability of the model. In this context, the initial assumptions of this study have been reconfirmed, the REM-robust regression data highlighting the same four factors (Share_Renew_En, Circ_Mat_Use, Tertiary_Education, and Gov_Effectiv) as effectively influencing the reduction of air pollution by particles, and a single factor that significantly influences the increase in air pollution by particles (Av_CO_2_).

In the REM-robust, Wald chi2(11) is 75.92, compared to the simple form of REM, where Wald chi2(11) was 174.34. At the same time, the materiality level of the Circ_Mat_Use factor has increased, with *p*-value in the REM-robust of 0.042 and |t_Student_| an improved value (2.03) as compared to the REM model, where *p*-value was 0.061 and |t_Student_| was lower (1.88).

This statistical data shows an improvement of the CE instrument to the REM-robust as compared to the basic form of REM.

## 5. Discussion

The results obtained reveal a negative relationship between the dependent variable and the main factors analyzed, so the most effective methods of reducing air pollution by particles were achieved by the use of renewable energy, increasing the circular material use rate, increasing the education of the population, and the government effectiveness, which has led to awareness of the situation and a pro-active attitude toward action to reduce environmental pollution, and to the proper implementation of the CE, as well as through responsible policies by EU governments.

It has also emerged that particular attention needs to be paid to factors that have a positive relationship with the response variable, represented by the increase in average urban income, which has also led to CO_2_-generating activities and significantly influenced the increase in air pollution by particles.

While it was expected that an increased share of environmental taxes in total taxes would curb or even reduce the level of pollution by taxing the economic activities that produce nuisances, this indicator did not prove to have the desired effect, bringing about the need to review the taxation levers for polluting activities. At the same time, investments in environmental protection need to be increased in order to be able to significantly influence the reduction of air pollution. All of these measures need to be combined in order to produce an ample effect to reach the desired results. To obtain sustainable increased urban incomes, they need to be based on activities with low levels of pollution. Otherwise, the level of penalties for polluting activities should be increased.

At the same time, people must be educated on targeted issues such as waste-reduction activities, recovery activities, and the reintroduction of reusable materials into the economy as much as possible.

When comparing the methods used and the results obtained in this paper with those of similar studies on sustainable economic development, there is obviously a common element, namely, the implementation of the CE principles, by “reduction-reuse-recycle”, which leads to a significant drop in the final waste. In this way, and as mentioned before, it is possible to obtain the much desired decrease in the consumption of natural (primary) resources, to protect the natural environment, and reduce pollution.

To pinpoint the importance of implementing and developing the CE, most researchers used comparative methods, obtaining different results, as highlighted in the following paragraphs.

For instance, in one paper, when examining the link between five CE indicators that included key components of economic and environmental growth, and GDP per capita using FEM and GMM analysis, the researchers identified a positive impact of almost all the CE variables on economic growth (except for the Trade in Recyclable Raw Materials indicator) [50]. As in our paper, the study was conducted for all 28 EU countries, but for a time frame that covered a larger period (2000–2017). 

A much closer approach to the one developed in the current paper in studying the dependencies between CE indicators in the EU is to be identified in another study examining a panel regression between 2010–2014. The researchers examined the progress of the CE by applying the fixed and random effects regression models and estimating the best model for establishing the dependency between main independent indicators (Circular Material Use Rate, R&D Expenditure by All Sectors, Resource Productivity and Domestic Material Consumption, Trade in Recyclable Raw Materials, Environmental Tax Revenues) and the dependent indicator (Recycling Rate of Municipal Waste). The tests employed brought about the conclusions that REM is a more suitable model than FEM, that the resource productivity and domestic material consumption has a strong statistical relationship with municipal waste recycling, and that an increase in the productivity and domestic material consumption is associated with a positive change in municipal waste recycling [53].

Case studies were also examined outside of Europe. A research paper conducted in China aimed to highlight the influence of three factors (Time, Districts/Counties, and Economic Development Level) on variations in MSW collection quantities, using a two-level hierarchical linear model (HLM) for 287 districts/regions, and a 10-year time interval (2008–2017). The analysis revealed a strong negative correlation between the level of regional economic development and the trend of increasing MSW collection quantities. Empirical findings indicate that the level of economic development and waste collection measures are critical elements of deterring MSW collection quantities to reduce environmental pollution [105].

The goal of the present research was to determine the influence of the main variables of the economic performance in urban areas, of the community development assistance activities, of the environmental protection activities, of the CE, and of the sustainable development and governance. These measures represent indicators of the CE implementation progress in Europe on the air pollution phenomenon. 

For a clearer picture of the evolution over time of the dependent variable that describes the exposure to air pollution, the specificities of each country and their evolution needed to be addressed. To analyze the heterogeneity factor, two models were considered for comparison in the OLS regression of the panel data: FEM, and REM. The Hausman test provided important information in deciding which model is more appropriate and would better support the proposals for future action to achieve the intended objective (reduction of air pollution), the REM alternative proving to be more suitable.

Under the 2021 State of Health in the EU country profile, experts from the Organization for Economic Cooperation and Development (OECD), and the European Observatory on Health Systems and Policies, for the 27 EU Member States, plus Finland and Iceland, air pollution has been identified as an important health risk factor. Air pollution has significant economic and human costs, increasing disease risk, reducing life expectancy, causing about 400,000 premature deaths per year among EU Member States, and, implicitly, reducing productivity [106]. A population affected by pollution is resource-intensive, and the costs of air pollution are found in rising costs for medical services and social services, with all economic activities being directly affected [107].

The costs of implementing measures to improve air quality have been estimated at 70–80 billion euro/year, but the cost of air pollution, for health, society and economic activities is much higher, estimated to be between 330 and 940 billion euro/year. A good policy to prevent and reduce air pollution can protect the working population and thus maintain its capacity to support the economic and social activities of the country.

With respect to the future research directions, the study might be extended by using criteria for the selection and efficient estimation of the generalized method of moments (GMM) for heteroskedastic models to estimate the minimum distance of parameters. In addition, a potential direction for future scientific papers could be that of performing more specific analyses to reveal the link between CE and LG and their impact on SD.

## 6. Conclusions

Increasing the potential and achieving a real CE can be implemented in urban areas in the following ways: by educating the population to make the proper use of the selective waste collection system; by a better re-use of resources in the local and national economy; by implementing a timetable of specific measures and tools; by developing online waste trading platforms to make production and/or distribution more efficient; and by creating an online database/stakeholder information portal on how to implement CE. All of these can help to achieve one of the objectives of smart city development, namely, to improve the quality of life of residents by ensuring an environmentally friendly environment.

So far, many EU Member States have not yet developed adequate waste management infrastructure. It is therefore important to set clear long-term policy objectives to guide measures and investments, to develop systems and mechanisms for waste treatment, and to block recyclable materials at the lower levels of the waste hierarchy.

Municipal waste management requires a very complex system, including an efficient collection scheme, an efficient sorting system and proper tracking of waste streams, the active involvement of citizens and businesses, the adaptation of infrastructure to the specific composition of waste, and a detailed financing system.

In recent times, city administrations are facing increasingly complex challenges to be able to implement and develop viable CE systems based on reliable and comparable data and to allow for more effective monitoring of progress toward the proposed objectives. This includes the need to create or redesign an appropriate infrastructure. It is also important to understand and create opportunities for new business models that support sustainable urban development. These should be achieved through collaborative actions between different stakeholder groups such as business, policy makers, NGOs, academia, etc.

Important steps can also be taken by using tools and measures for the recovery of reusable materials as much as possible, by reducing final waste, thereby contributing significantly to the reduction of environmental pollution by applying the provisions of Article 4(3), Annex IV of European Parliament Directive No 851/2018 on waste. 

These instruments and measures are based on: the application of taxes and limitations on landfilling of waste and incineration of waste; taxation of the actual amount of waste generated; economic incentives for local and regional authorities to promote waste generation limitation and the intensification of separate collection schemes; good planning of investments in waste management infrastructure; research and innovation in advanced waste treatment, recycling and remanufacturing technologies; public education and awareness campaigns to reduce environmental pollution by garbage; and stimulation of the selective collection of waste.

The purpose of this study was to highlight the influence of circular economy-specific indicators, fiscal levers, growth-specific indicators, and environmental indicators on how to achieve an important objective of sustainable urban development, namely, ensuring a healthy environment, with as little pollution as possible.

The indicators considered in the analysis are only a part of the means by which urban authorities could take the necessary measures to encourage activities that help reduce waste as much as possible, and reduce the harmful factors that increase air pollution. The aim of this paper was also to use a balanced nine-year panel data set. Unfortunately, for many indicators the latest data available on Eurostat only covered the period up to 2019. It is important to point out that, although the intention was to address a much longer time frame, we are not certain that the relevance of the results achieved would have increased because the economic, social, and international political phenomena, recently, have evolved rapidly, and it is important that such studies are constantly updated.

The results of this study could help and provide useful information to the development of mechanisms and the promotion of policies to achieve two of the objectives of SD, Goal 3—Good health and well-being, and Goal 11—Sustainable cities and communities.

## Figures and Tables

**Figure 1 ijerph-19-07627-f001:**
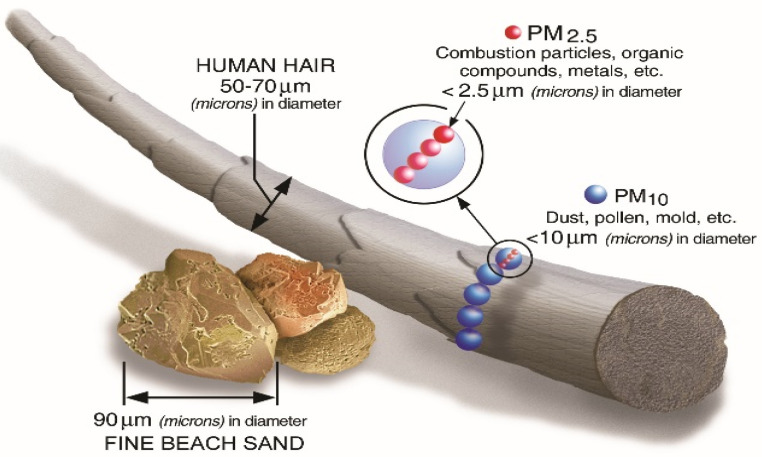
Size comparisons for PM particles. Source: retrieved from The United States Environmental Protection Agency (EPA) official website.

**Figure 2 ijerph-19-07627-f002:**
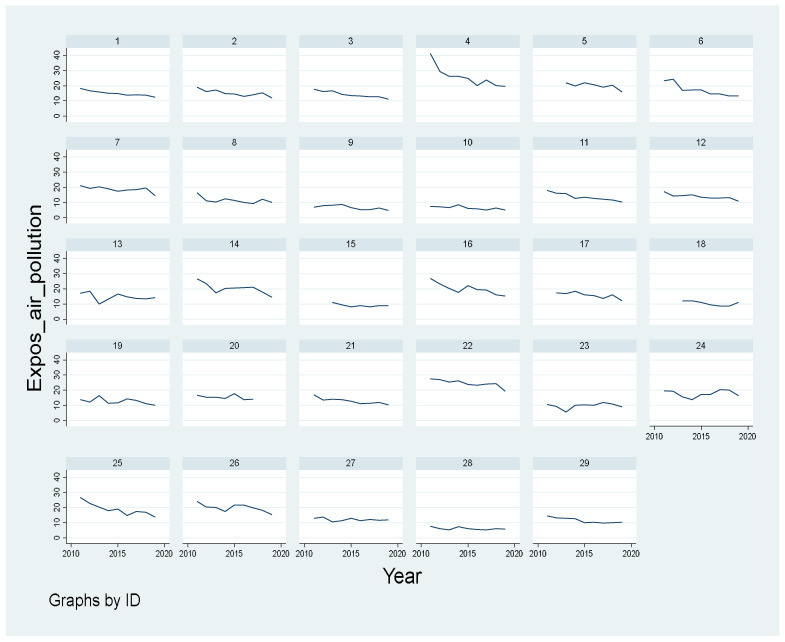
Exposure to air pollution by particulate matter (2011–2019). Source: authors’ representation of the data in Stata graph.

**Table 1 ijerph-19-07627-t001:** The independent variables used for the panel data regression of the EU-28 Member States.

Sustainable Development Objective	Specific Indicators	Abbreviation	Unit of Measurement	Description/Relevance
1. Economic performance of urban areas and support activities for community development	1.1. Mean income by degree of urbanization	1.1. Mn_Urb_Inc	1.1. Euro per year	1.1. Living conditions indicator
	1.2. Official development assistance	1.2. Develop_Assist	1.2. Share of Gross Net Income (% of GNI)	1.2. Grants or loans that are undertaken by the official sector with the objectives of promoting economic development and welfare in the recipient countries
2. Environmental protection activities	2.1. Share of environmental taxes	2.1. Share_Env_Tax	2.1. Percentage of environmental taxes into the total tax revenues (% of total taxes and social contributions)	2.1. Taxes whose tax base is a physical unit (or proxy of it) of something that has a specific negative impact on the environment; environmental taxes are implemented for those activities that have a negative impact on the environment; the tax revenues stem from four types of taxes: energy taxes *, transport taxes **, pollution taxes and resource taxes ***
	2.2. Environmental protection investments of total economy	2.2. Env_Prot_Inv	2.2. Million euro	2.2. The economic resources allocated to activities and actions to prevent, reduce and eliminate environmental pollution (air, water, soil, and noise) and any other degradation of the environment; the environmental protection expenditure related to waste water management, biodiversity protection, R&D, education and training contribute directly to the EU’s policy priorities on environmental protection, resource management and green growth by providing information on the production and the use of environmental protection services.
	2.3. Average carbon dioxide	2.3. Av_CO_2_	2.3. Grams of carbon dioxide per km	2.3. The emissions per km by new passenger cars in a given year; the indicator is covered by three European sustainable development objectives (SDO) ****
3. Circular economy	3.1. Trade in recyclable raw materials by waste	3.1. Trade_Rec_Waste	3.1. Thousand euro	3.1. Imports of recyclable raw materials by waste
	3.2. Recycling rate of municipal waste	3.2. Rec_Rate_Mun_Waste	3.2. Thousand tons	3.2. The share of recycled municipal waste in the total municipal waste generation. It covers the objective of sustainable cities and communities
	3.3. Circular material use rate	3.3. Circ_Mat_Use	3.3. Percentage (%)	3.3. Responsible consumption and production
4. Sustainable development	4.1. Share of renewable energy in gross final energy consumption by sector	4.1. Share_Renew_En	4.1. Percentage (%)	4.1. Its development is being monitored and covered by two European SDO: affordable and clean energy and climate action
	4.2. Tertiary educational attainment	4.2. Tertiary_Education	4.2. Percentage (%)	4.2. Measures the share of the population aged 25–34 who have successfully completed tertiary studies; its development is monitored and covered by two European SDO: quality education, and gender equality
5. Governance	5.1. Government Effectiveness	5.1. Gov_Effectiv	5.1. Ranges from −2.5 (weak governance performance) to 2.5 (strong governance performance)	5.1. Captures perceptions of the quality of public services, and the degree of its independence from political pressures, the quality of policy implementation, and the credibility of the government’s commitment to such policies; countries are ranked according to percentage score ranging from 0-lowest—to 100-highest

* The energy taxes contribute around 75% of the total environmental taxes; ** the transport taxes contribute about 20% of the total environmental taxes; *** the pollution and resource taxes represent about 5% of the total environmental taxes; **** industry, innovation and infrastructure; responsible consumption and production; climate action.

**Table 2 ijerph-19-07627-t002:** The dependent variable used for the panel data regression of the EU-28 Member States.

Response Variable	Specific Indicator	Abbreviation	Unit of Measurement	Description/Relevance
Air pollution	Exposure to air pollution by particulate matter (particulates < 2.5 μm)	Expos_air_pollution	Micrometer (µm)	Measures the weighted average annual concentration of suspended particles, smaller than 2.5 μm, in urban base stations in agglomerations; fine particles < 2.5 µm (PM2.5) are a subset of gross particles micrometers of 10 (PM10), PM2.5 harmful health impact being more serious than PM10 because they can be pulled further into the lungs. PM2.5 are very toxic as they are transported deep into the lungs, where they can cause inflammation and exacerbate the condition of people suffering from heart and lung diseases

**Table 3 ijerph-19-07627-t003:** Descriptive statistics for total EU28 sample.

Variable	Observations	Mean	Standard Deviation	Minimum	Maximum
Expos_air_pollution	252	14.75	5.61	4.80	41.30
Mn_Urb_Inc	252	17,596.43	9930.46	2936	48,452
Develop_Assist	252	0.33	0.28	0.04	1.40
Share_Env_Tax	252	7.45	1.75	4.32	11.75
Env_Prot_Inv	252	2214.10	3188.53	11.90	13,124.90
Av_CO_2_	252	125.28	11.30	98.40	156.90
Trade_Rec_Waste	252	1,224,520	1,959,527	168	10,342,858
Rec_Rate_Mun_Waste	252	34.99	15.31	7.40	67.20
Circ_Mat_Use	252	8.81	6.31	1.30	30
Share_Renew_En	252	19.52	11.63	1.85	56.39
Tertiary_education	252	39.34	8.89	20.90	60.30
Gov_Effectiv	252	81.91	12.10	45.19	100

Source: authors’ calculation based on Eurostat and The World Bank’s databases.

**Table 4 ijerph-19-07627-t004:** Exposure to air pollution by particulate matter dynamics in 2019 as compared to 2011.

ID. Country	Air Pollution 2011 (µm)	Air Pollution 2019 (µm)	Air Pollution Reduction 2019/2011 (%)
1. Average EU28	18.40	12.60	31.52
2. Austria	19.00	12.00	36.84
3. Belgium	17.70	11.10	37.29
4. Bulgaria	41.30	19.60	52.54
5. Croatia	21.90	16.00	26.94
6. Cyprus	23.20	13.40	42.24
7. Czech Republic	21.00	14.40	31.43
8. Denmark	16.30	10.00	38.65
9. Estonia	6.90	4.80	30.43
10. Finland	7.50	5.10	32.00
11. France	17.80	10.40	41.57
12. Germany	17.10	10.90	36.26
13. Greece	17.00	14.10	17.06
14. Hungary	26.50	14.40	45.66
15. Ireland	11.10	8.80	20.72
16. Italy	26.80	15.10	43.66
17. Latvia	17.30	12.10	30.06
18. Lithuania	12.10	11.10	8.26
19. Luxembourg	13.70	10.20	25.55
20. Malta	16.60	13.90	16.41
21. The Netherlands	16.80	10.40	38.10
22. Poland	27.60	19.30	30.07
23. Portugal	10.70	9.10	14.95
24. Romania	19.50	16.40	15.90
25. Slovakia	26.70	13.80	48.31
26. Slovenia	24.10	15.30	36.51
27. Spain	12.90	11.80	8.53
28. Sweden	7.80	5.80	25.64
29. UK	14.60	10.20	30.14

Source: authors’ calculation based on Eurostat database.

**Table 5 ijerph-19-07627-t005:** The influence of the independent indicators on the target variable.

Independent Variable	*t*-Stat Value	*p* > |*t*|	Real Influence on the Response Variable	Ideal Influence on the Response Variable
Mn_Urb_Inc	1.43	0.155	Weak link/+	Strong/−
Develop_Assist	−2.74	0.007 ***	Strong/−	Strong/−
Share_Env_Tax	3.81	0.000 ***	Strong/+	Strong/+
Env_Prot_Inv	1.58	0.116	Weak link/+	Strong/−
Av_CO_2_	7.27	0.000 ***	Strong/+	Strong/+
Trade_Rec_Waste	−3.11	0.002 ***	Strong/−	Strong/−
Rec_Rate_Mun_Waste	3.11	0.002 ***	Strong/+	Strong/−
Circ_Mat_Use	0.59	0.557	Weak link/+	Strong/−
Share_Renew_En	−6.31	0.000 ***	Strong/−	Strong/−
Tertiary_Education	−5.13	0.000 ***	Strong/−	Strong/−
Gov_Effectiv	−4.44	0.000 ***	Strong/−	Strong/−

1 * 00iry_e.d in the main text (page n explained and the results more adequatenk between them expressed by rho.bles.en the unrest). A *p*-value is statistically significant if: *p* < 0.01 ***. Source: authors’ calculation using an econometric software.

**Table 6 ijerph-19-07627-t006:** The countries with the highest/lowest values in terms of the quantity of air pollution.

Factors Which May Influence the Decrease in Air Pollution	Unit of Measurement	Annual EU Average (2011–2019)	Countries with the Highest Values	Countries with the Lowest Values
Share_Renew_En	Percentage	16.41	Sweden (52.47)	Luxembourg (5.17)
			Finland (38.45)	Malta (5.36)
			Latvia (37.73)	The Netherlands (5.92)
			Austria (33.11)	UK (7.98)
Tertiary_Education	Percentage	37.73	Cyprus (55.13)	Italy (24.87)
			Ireland (53.22)	Germany (30.02)
			Lithuania (52.89)	Czech Republic (30.59)
			Luxembourg (51.15)	Bulgaria (31.11)
Gov_Effectiv	Percentage	81.91	Finland (99.36)	Romania (52.05)
			Denmark (97.71)	
			Sweden (95.68)	Bulgaria (59.39)
			The Netherlands (95.16)	Greece (63.85)
			Luxembourg (94.52)	Italy (69.49)
Trade_Rec_Waste	Thousand euro/year	1,224,520	Germany (9,107,102)	Cyprus (399)
			Italy (4,208,801)	Malta (460)
			Belgium (4,155,033)	Croatia (41,224)
			Spain (3,062,393)	Estonia (56,787)
				Ireland (77,073)
Develop_Assist	Percentage from GNI	0.42	Sweden (1.05)	Latvia (0.09)
			Luxembourg (0.99)	Croatia (0.09)
			Denmark (0.79)	Romania (0.09)
			UK (0.67)	
			The Netherlands (0.66)	Bulgaria (0.10)
			Germany (0.52)	

Source: authors’ calculation using an econometric software.

**Table 7 ijerph-19-07627-t007:** Statistical significance of the considered variables.

Independent Variables/Indicators	Pooled OLS *p* > |*t*|	FEM *p* > |*t*|	REM *p* > |*t*|
Mn_Urb_Inc	0.155	0.014	0.213
Develop_Assist	0.007	0.822	0.714
Share_Env_Tax	0.000	0.136	0.102
Env_Prot_Inv	0.116	0.605	0.408
Av_CO_2_	0.000	0.001	0.000
Trade_Rec_Waste	0.002	0.440	0.332
Rec_Rate_Mun_Waste	0.002	0.555	0.696
Circ_Mat_Use	0.557	0.052	0.061
Share_Renew_En	0.000	0.001	0.000
Tertiary_Education	0.000	0.251	0.007
Gov_Effectiv	0.000	0.148	0.004
Constant	0.037	0.335	0.067
Number of observations	177	177	177
Number of groups	27	27	27
F-statistic (11, 171)/Wald chi(11)	30.91	14.42	174.37
Prob. > F/Prob > chi2	0.0000	0.0000	0.0000
R-Squared-within	0.6733	0.5330	0.5029
R-Squared-between		0.1275	0.5792
R-Squared-overall		0.1614	0.5857
Sigma_u		6.8828346	3.8287066
Sigma_e		1.7280559	1.7280559
rho		0.94070291	0.83076519

Source: authors’ calculation using an econometric software.

**Table 8 ijerph-19-07627-t008:** The statistical significance of the factors and the coefficients for FEM-robust and REM-robust.

Explanatory Variables	FEM Robust				REM Robust			
	Coefficient	Standard Error	*t*	*p* > |*t*|	Coefficient	Standard Error	*t*	*p* > |*t*|
Mn_Urb_Inc	0.0004	0.0001775	2.02	0.054 *	0.0001	0.0000918	1.25	0.211
Develop_Assist	0.6128	1.515889	0.40	0.689	−0.8488	1.517565	−0.56	0.576
Share_Env_Tax	0.6142	0.6565595	0.94	0.358	0.5268	0.4539645	1.16	0.246
Env_Prot_Inv	0.0002	0.0002676	0.64	0.530	0.0002	0.0002374	0.84	0.403
Av_CO_2_	0.1424	0.0741921	1.92	0.066 *	0.1628	0.060116	2.71	0.007 ***
Trade_Rec_Waste	−0.0000004	0.00000047	−0.83	0.413	−0.0000004	0.000000351	−1.04	0.298
Rec_Rate_Mun_Waste	−0.0218	0.043798	−0.50	0.622	0.0131	0.0503287	0.26	0.795
Circ_Mat_Use	−0.2151	0.1032436	−2.08	0.047 **	−0.1695	0.0835412	−2.03	0.042 **
Share_Renew_En	−0.5497	0.174055	−3.16	0.004 ***	−0.2387	0.0605946	−3.94	0.000 ***
Tertiary_Education	−0.0939	0.0826589	−1.14	0.266	−0.1629	0.0672452	−2.42	0.015 **
Gov_Effectiv	−0.0873	0.0693172	−1.26	0.219	−0.1450	0.0638089	−2.27	0.023 **
Constant	10.9877	16.32887	0.67	0.507	13.1079	9.967409	1.32	0.188
Number of observations	177				177			
Number of groups	27				27			
F-statistic (11, 26)/Wald chi(11)	18.80				75.92			
Prob. > F/Prob > chi2	0.00				0.00			
R-Squared-within	0.5330				0.5029			
R-Squared-between	0.1275				0.5792			
R-Squared-overall	0.1614				0.5857			
Sigma_u	6.8828346				3.8287066			
Sigma_e	1.7280559				1.7280559			
rho	0.94070291				0.83076519			

A *p*-value is statistically significant if: *p* < 0.01 ***, *p* < 0.05 **, *p* < 0.10 *. Source: authors’ calculation using an econometric software.

## Data Availability

The initial data was processed by authors using the Eurostat and The World Bank databases. The data presented in this study are available on request from the corresponding author.

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
