# Peer review of "Does Circular Economy Contribute to Smart Cities’ Sustainable Development?"

_ijerph, 2022, doi:10.3390/ijerph19137627_

Round 1

Reviewer 1 Report

The subject of the article is very interesting and topical. It is adequate and relevant to the journal theme and scope. It deals with critical issues related to the development of the city, its impact on the environment and the reduction of this impact.

I hope this paper will be published, however, I suggest considering following improvements:

- The subject of the article is the question of whether circular economy contribute to smart cities’ sustainable development. It is not clear whether the authors assumed that the studied cities are smart cities (as indicated by the title of the article), or as briefly described in the summary, they are looking for smart solutions supporting the reduction of the negative impact on the environment of cities. The concept of smart city appears in the introduction, and only later in the summary. It does not appear in the objective or methodology of the research carried out. I suggest that the title be made more specific or that the specificity of the smart city concept should be included in the description of the research methodology and objective. Please also note that the following concepts are discussed in the literature: smart city, sustainable city, and smart and sustainable city.

- In Table 2 containing the description of the dependent variable (Air pollution). It is not clear from the description what the suspended particles concerns. Only fine particles <2.5 μm? Is the unit of measurement for the average weighted annual concentration of suspended particles in urban base stations in agglomerations a micrometre (μm)?

- The regression equation contains indices that are not adequately described. What do the indices "i" and "t" and the parameters "α", "μi" and "εit" mean? It may not be obvious to the reader that Coefficient (Table 8) is the B value from the equation (1).

- Please explain / clarify the symbols used in the tables in the text of the article.

- Some coefficient values have small values. I believe that authors should also present the values of their standard errors.

I would like to thank the authors for their work in developing the article and see this manuscript published but considering at least the above issues.

Good luck.

Author Response

Response to Reviewer 1 Comments

Comments on the reviewer’s report for the paper:

 Does Circular Economy Contribute to Smart Cities’ Sustainable Development?

            Our comments refer to the suggestions and recommendations the referees made in order to improve the quality of the research paper, and not to the positive remarks they made. We have to mention that most requirements were fulfilled in this new version of the paper. The pages and line numbers mentioned between brackets for the changes we applied to answer the reviewer’s suggestions refer to the new version of the paper.

Dear Reviewer,

            First of all, we would like to express our gratitude for the work you dedicated for investigating our research, identifying the points for improvement, and suggesting ways for achieving that. We are fully aware that your suggestions and recommendations are very important for improving our research, and the way it is presented in this article.

            Thank you and we hope we have answered all your suggestions and recommendations and improved our research.

Comment:

            1) “The subject of the article is the question of whether circular economy contribute to smart cities’ sustainable development. It is not clear whether the authors assumed that the studied cities are smart cities (as indicated by the title of the article), or as briefly described in the summary, they are looking for smart solutions supporting the reduction of the negative impact on the environment of cities. The concept of smart city appears in the introduction, and only later in the summary. It does not appear in the objective or methodology of the research carried out. I suggest that the title be made more specific or that the specificity of the smart city concept should be included in the description of the research methodology and objective. Please also note that the following concepts are discussed in the literature: smart city, sustainable city, and smart and sustainable city.”

Response

Thank you very much for this observation. We started from the premise of conducting a research on the development of smart cities. However, we must refer to an obvious reality, namely to the fact that cities are constantly changing and adapting, and the objective of "smart environment" is permanently under pressure, as a result of the social interactions and economic activities that characterize the urban population and the business class. As a consequence, the circular economy (CE) is important for all processes and activities carried out in the urban environment. Moreover, we need to take into account several factors and their combination in applying mechanisms for developing and supporting a smart environment.

The suggested changes have been made in the abstract (p. 1, line 12-17) and in the third part of the paper referring to the materials and methods (pp. 9-10, line 435-475). Several significant references were added to support the idea of a strong relationship between CE and smart cities (reference 100, p. 9, line 449; reference 101, p. 10, line 456; reference 102, p. 10, line 456 and 475).

Comment:

2) “In Table 2 containing the description of the dependent variable (Air pollution). It is not clear from the description what the suspended particles concerns. Only fine particles <2.5 μm? Is the unit of measurement for the average weighted annual concentration of suspended particles in urban base stations in agglomerations a micrometre (μm)?”

Response

The indicator refers only to the fine particulates (PM2.5), i.e. particulates whose diameters are less than 2.5 micrometers, which are a subset of the fine and coarse particulates of 10 micrometers diameter (PM10). PM2.5 deleterious health impacts are more serious than PM10 as they can be drawn further into the lungs and may be more toxic. All these explanations were introduced in Table 2 (p. 12, line 504, columns 2 and 5).

            Furthermore, to better understand the difference between the PM particles, we introduced a new image (Figure 1 on p. 12, line 511-513) and several arguments and explanations related to it (p. 12, line 505-517).

Comment:

      3) “The regression equation contains indices that are not adequately described. What do the indices "i" and "t" and the parameters "α", "μi" and "εit" mean? It may not be obvious to the reader that Coefficient (Table 8) is the B value from the equation (1).”

Response

We have completed this part by offering a legend for the indices that appear in the equation. The indices introduced in equation 1 are now described properly on page 13, line 520-530.

Comment:

     4) “Please explain / clarify the symbols used in the tables in the text of the article.”

Response

            All the symbols used in the tables for all the variables, as well as in the main text of the paper, are clearly presented in Table 1 (p. 10-11, line 498-503), and Table 2 (p. 12, line 504), on the third column. The mentioned tables present the 11 variables used for the panel data analysis as specific indicators representing several sustainable development objectives (SDOs), and the dependent variable. Besides the main objective and the abbreviation of each indicator, the unit of measurement, the relevance, and a brief description are also available. We consider that it would be redundant to mention once again the symbols of the 12 variables.

            Some other symbols identified in the tables are:

- “min” and “max” on Table 3, which were replaced by “minimum” and “maximum” (p. 14, line 535);

- “EU” and “UK” on Table 4 and Table 6; we added the symbol for the European Union when it was first used in the text (p. 2, line 84), while the symbol of the United Kingdom was properly introduced when the country was first mentioned in the paper (p. 4, line 172-173);

- “t-Stat” and “p>|t|” values from Table 5 and Table 8 for which we added some interpretations and explanations before and after Table 5 (p. 17, line 641-648 and p. 18, line 654-657);

- “OLS” on Table 7 is first introduced on page 6, line 252; some additional information about OLS is to be found on page 15, line 578-581;

- “FEM” and “REM” on Table 7; FEM is abbreviated for the first time on page 5, line 235, while REM is explained on page 19, line 690;

- “No.” from Table 7 and Table 8 were replaced by “Number” (p. 19, line 700, respectively p. 21, line 765);

- “F-statistic”, “Wald chi”, “R-Squared”, “Sigma_u”, “Sigma_e” and “rho” seen in Table 7 and Table 8 are also explained and their results for the models adopted for comparison more adequately presented in the main text (p. 19, line 695-699, respectively p. 20, line 702-713).

Comment:

 5) “Some coefficient values have small values. I believe that authors should also present the values of their standard errors.”

Response

Thank you very much for this recommendation. We added the standard errors on a separate column of Table 8 (p. 21, line 764-765).

Best Regards,

The authors

Reviewer 2 Report

The Abstract is well contextualised, however it is very vague, and should be improved by the authors. It would be very good if the authors indicated which Research Methodology was used in this research, as well as indicating in the Abstract some of the strongest values obtained by the authors in the conclusions.

[line 40] The authors, quite rightly, make a very interesting statement, which allows us to give great relevance to this research: “The CE is seen as a sustainable economic system where economic growth…” - However, without a scientific reference to back it up, it makes one unfortunately question this statement. I recommend the authors to attach a recent scientific reference.

[line 43-45] The authors make a very interesting connection, between the Circular Economy, and the reality of today's businesses: “It is a regenerative system in which resource input and waste, emission, and energy leakage are minimized by slowing, closing, and narrowing energy and material loops.” - To give a greater scope and importance to the article, it would be important at this point for the authors to refer to the "Lean Green", which is related to the concern of organizations that adopt the lean philosophy, in measuring the benefits of these practices in environmental terms. As a suggestion to the authors, among other articles I can recommend the article: "Lean Green—The Importance of Integrating Environment into Lean Philosophy—A Case Study" (https://doi.org/10.1007/978-3-030-41429-0_21).

[line 88-90] The authors make the following statement throughout the article: - “The term of CE has become more and more complex, by incorporating many concepts that focus on sustainability, such as: industrial ecology, eco-efficiency, waste management, renewable energy, recycling, smart cities.” - What is the relationship between "Circular Economy" and " Smart Cities "? It is not clear this passage between these two concepts for future readers. It would be important if the authors were able to clarify this, and if possible, add a reference, to support this statement.

[line 102] The statement: “In a recent paper identified in the literature, there is a rather critical analysis of the…”- It doesn't make much sense, because what would make this stronger, would be to indicate the author or authors who did the critique. For example, authors putting: "Friant et al., did a rather critical analysis of the 102 EU’s discourse followed by a complex...".

[line 144] By addressing here something very important in the theme of this research as the "Triple Bottom Line", it would be very important that the authors leave here some associated scientific references, which would allow future readers and the scientific community to better understand this theme, and more easily realize the importance of this issue. As a suggestion to the authors, among other articles I can recommend the article: " Combining lean and green practices to achieve a superior performance: The contribution for a sustainable development and competitiveness—An empirical study on the Portuguese context" (https://doi.org/10.1002/csr.2242).

[line342] It would be very important for the purpose of this research, if the authors in the topic: "3. Materials and Methods", had identified their "Research Question", which supports the pertinence of this research!

The authors deme in the abstract to recommend future work. It would be very important that other researchers in the future could develop research that could give continuity to the research developed by the authors. This type of information is very important for the scientific community.

Author Response

Response to Reviewer 2 Comments

Comments on the reviewer’s report for the paper:

 Does Circular Economy Contribute to Smart Cities’ Sustainable Development?

            Our comments refer to the suggestions and recommendations the referees made in order to improve the quality of the research paper, and not to the positive remarks they made. We have to mention that most requirements were fulfilled in this new version of the paper. The pages and line numbers mentioned between brackets for the changes we applied to answer the reviewer’s suggestions refer to the new version of the paper.

Dear Reviewer,

            First of all, we would like to express our gratitude for the work you dedicated for investigating our research, identifying the points for improvement, and suggesting ways for achieving that. We are fully aware that your suggestions and recommendations are very important for improving our research, and the way it is presented in this article.

            Thank you and we hope we have answered all your suggestions and recommendations and improved our research.

Comment:

1) “The Abstract is well contextualised, however it is very vague, and should be improved by the authors. It would be very good if the authors indicated which Research Methodology was used in this research, as well as indicating in the Abstract some of the strongest values obtained by the authors in the conclusions.”

Response

Thank you very much for your observation! We have completed the research methodology by mentioning also the other two statistical models referring to the Fixed Effects and the Random Effects (p. 1, line 17-19). The abstract was reorganised and the strongest values obtained in our paper were better pinpointed (p. 1, line 20-29).

Comment:

2) “[line 40] The authors, quite rightly, make a very interesting statement, which allows us to give great relevance to this research: “The CE is seen as a sustainable economic system where economic growth…” - However, without a scientific reference to back it up, it makes one unfortunately question this statement. I recommend the authors to attach a recent scientific reference.”

Response

We fulfilled this recommendation (p. 2, line 47-56) by adding three more references to this part, to scientifically support the strong connection between the circular economy (CE) and the sustainable development (goals) (SD(Gs)): reference 3 (p. 2, line 50-52), reference 4 (p. 2, line 52-54), and reference 5 (p. 2, line 54-56).

Comment:

      3) “[line 43-45] The authors make a very interesting connection, between the Circular Economy, and the reality of today's businesses: “It is a regenerative system in which resource input and waste, emission, and energy leakage are minimized by slowing, closing, and narrowing energy and material loops.” - To give a greater scope and importance to the article, it would be important at this point for the authors to refer to the "Lean Green", which is related to the concern of organizations that adopt the lean philosophy, in measuring the benefits of these practices in environmental terms. As a suggestion to the authors, among other articles I can recommend the article: "Lean Green—The Importance of Integrating Environment into Lean Philosophy—A Case Study" (https://doi.org/10.1007/978-3-030-41429-0_21).”

Response

Your remark is a very good one as the “Lean Green” (LG) philosophy aims to produce developments that lower the demand for resources, provide efficient structures, and deploy innovative technology. It is a concept very much connected to the CE and SD.

            We have completed the paragraph you mentioned according to several relevant research papers found in the literature, including the one you recommended. The information we added and the specific literature overview (references 6-10) are to be found on page 2, line 61-73. Obviously, a few phrases do not cover entirely all the implications of such an important concept, but our intention is to conduct a rigorous analysis and reveal the link between CE, SDGs and LG in a future paper.

  Comment:

4) “[line 88-90] The authors make the following statement throughout the article: - “The term of CE has become more and more complex, by incorporating many concepts that focus on sustainability, such as: industrial ecology, eco-efficiency, waste management, renewable energy, recycling, smart cities.” - What is the relationship between "Circular Economy" and " Smart Cities "? It is not clear this passage between these two concepts for future readers. It would be important if the authors were able to clarify this, and if possible, add a reference, to support this statement.”

Response

Thank you very much for this observation. We started from the premise of conducting a research on the development of smart cities. However, we must refer to an obvious reality, namely to the fact that cities are constantly changing and adapting, and the objective of "smart environment" is permanently under pressure, as a result of the social interactions and economic activities that characterize the urban population and the business class. As a consequence, the circular economy (CE) is important for all processes and activities carried out in the urban environment. Moreover, we need to take into account several factors and their combination in applying mechanisms for developing and supporting a smart environment.

We clarified and made a few improvements in our paper to better underline these aspects. The changes have been made in the abstract (p. 1, line 12-17) and in the third part of the paper referring to the materials and methods (pp. 9-10, line 435-475). Several significant references were added to support the idea of a strong relationship between CE and smart cities: reference 101 (pp. 9-10, line 454-463) and reference 102 (pp. 9-10, line 454-456, 464-475).

Comment:

5) “[line 102] The statement: “In a recent paper identified in the literature, there is a rather critical analysis of the…”- It doesn't make much sense, because what would make this stronger, would be to indicate the author or authors who did the critique. For example, authors putting: "Friant et al., did a rather critical analysis of the 102 EU’s discourse followed by a complex...”

Response

Thank you. The way we introduced that paper was a little bit too vague. We made the necessary changes for the phrase to make much more sense, respecting at the same time the instructions available for authors (p. 3, line 137-142).

Comment:

6) “[line 144] By addressing here something very important in the theme of this research as the "Triple Bottom Line", it would be very important that the authors leave here some associated scientific references, which would allow future readers and the scientific community to better understand this theme, and more easily realize the importance of this issue. As a suggestion to the authors, among other articles I can recommend the article: " Combining lean and green practices to achieve a superior performance: The contribution for a sustainable development and competitiveness—An empirical study on the Portuguese context" (https://doi.org/10.1002/csr.2242).”

Response

            Thank you so much for your suggestion. The Triple Bottom Line (TBL) is one of the main systems being used by businesses to assess the profits they are making through their corporate sustainability solutions. This method asks us to see beyond the traditional bottom line of business to the profits that the business makes socially, environmentally, and economically. 8 references (references 39-46) were added to support these ideas (pp. 4-5, line 182-204).

  Comment:

  7) “[line342] It would be very important for the purpose of this research, if the authors in the topic: "3. Materials and Methods", had identified their "Research Question", which supports the pertinence of this research!”

Response

We have succeeded in better highlighting the research question we addressed even from the beginning also in the topic referring to the materials and methods used. Do the circular economy-specific indicators, fiscal levers, growth-specific indicators, and environmental indicators influence one important objective of sustainable urban development, namely ensuring a healthier environment, with as little pollution as possible? (p. 9, line 429-430).

Comment:

8) “The authors deme in the abstract to recommend future work. It would be very important that other researchers in the future could develop research that could give continuity to the research developed by the authors. This type of information is very important for the scientific community.”

Response

We are very much aware about the great significance of offering some directions for further analysis, and we did express our opinion in this respect in the end of the Discussion section (p. 24, line 882-886), as the template available for authors require: “Authors should discuss the results and how they can be interpreted … Future research directions may also be highlighted.”

We consider that this research might be extended by using criteria for estimating the minimum distance of parameters by GMM methodology for heteroskedastic models, and by performing more specific analyzes to reveal the link between CE and LG and their impact on SD.

Unfortunately, the abstract is limited to only 200 words, and, according to the instructions for authors provided on the journal’s website page, it should cover the background/purpose of the study, the main methods, the results/main findings, and a brief conclusion (https://www.mdpi.com/journal/ijerph/instructions). As already mentioned in observation 1, we have reorganized the abstract by better underlying the research methodology, and more clearly stating the strongest values obtained in our paper.

Best Regards,

The authors

Round 2

Reviewer 1 Report

I accept the changes made by the authors. The final layout of the content, the form of presentation, description and inference reflect the formal requirements. The article will certainly be interesting for readers.

I thank the authors for their work in developing the article and I wish them good luck.